# Nasopharyngeal Carriage, Serotype Distribution, and Antimicrobial Susceptibility of *Streptococcus pneumoniae* Among PCV13-Vaccinated and -Unvaccinated Children in Iran

**DOI:** 10.3390/vaccines13070707

**Published:** 2025-06-29

**Authors:** Fatemeh Ashrafian, Mona Sadat Larijani, Saiedeh Haji Maghsoudi, Delaram Doroud, Alireza Fahimzad, Zahra Pournasiri, Elham Jafari, Masoumeh Parzadeh, Sara Abdollahi, Elham Haj Agha Gholizadeh Khiavi, Anahita Bavand, Morvarid Shafiei, Mahdi Rohani, Amitis Ramezani

**Affiliations:** 1Clinical Research Department, Pasteur Institute of Iran, No: 69, Pasteur Ave, Tehran 1316943551, Iran; 2Modeling in Health Research Center, Institute for Futures Studies in Health, Kerman University of Medical Sciences, Kerman 7616913555, Iran; sa.maghsoudi@gmail.com; 3Department of Biostatistics and Epidemiology, School of Public Health, Kerman University of Medical Sciences, Kerman 7616913555, Iran; 4Department of Immunotherapy and Leishmania Vaccine Research, Pasteur Institute of Iran, Tehran 1316943551, Iran; delaramdoroud@yahoo.com; 5Division of Pediatric Infectious Diseases, Department of Pediatrics, Pediatric Infectious Research Center, Mofid Children Hospital, Shahid Beheshti University of Medical Sciences, Tehran 1546815514, Iran; safahimzad@yahoo.com; 6Pediatric Nephrology Research Center, Research Institute for Children’s Health, Shahid Beheshti University of Medical Sciences, Tehran 1546815514, Iran; pournasiri.z@gmail.com; 7Faculty of Converging Science and Technology, Islamic Azad University, Tehran 1811694784, Iran; elhamjafari988@gmail.com; 8Department of Bacteriology, Pasteur Institute of Iran, Tehran 1316943551, Iranelham.gholizade98@gmail.com (E.H.A.G.K.);

**Keywords:** *Streptococcus pneumoniae*, nasopharyngeal carriage, pneumococcal conjugated vaccines, *cpsB*

## Abstract

**Background and Aim**: Pneumococcal pneumonia is a major cause of death globally, emphasizing the importance of vaccination, especially in low- and middle-income countries. In Iran, the 13-valent pneumococcal conjugate vaccine (PCV13) is available exclusively through private healthcare systems, resulting in a lack of studies on the prevalence of *Streptococcus pneumoniae* (*S. pneumoniae*) serotypes among vaccinated children. This research aimed to explore and compare the prevalence of nasopharyngeal pneumococcal carriage, serotype distribution, and antibiotic resistance patterns in healthy PCV13-vaccinated and -unvaccinated children. **Methods**: From August 2023 to November 2024, a multi-center, cross-sectional observational study was conducted in Tehran, Iran. This study included 204 nasopharyngeal samples collected from children aged from 18 to 59 months, involving both cases of children vaccinated with PCV13 and unvaccinated populations. *S. pneumoniae* was identified through a combination of culture methods and biochemical tests, confirmed by real-time PCR. Serotyping was achieved using *cpsB* sequencing, and the minimum inhibitory concentration method was employed to assess antibiotic resistance. **Results**: This study revealed similar *S. pneumoniae* carriage rates between PCV13-vaccinated and -unvaccinated Iranian children (20.6% vs. 21.6%). Serotypes 23F and 19F were prevalent in unvaccinated children, while 15B/15C was more prevalent in PCV13-vaccinated children. The included *S. pneumoniae* serotypes in PCV13 were detected more in the unvaccinated group. PCV13-vaccinated children exhibited no penicillin-resistant pneumococcal isolates, although four isolates were non-susceptible in unvaccinated children. Both groups showed substantial resistance to erythromycin and SXT. Previous respiratory infections, daycare attendance, residence in Tehran, and a history of antibiotic consumption increased the risk of pneumococcal carriage. **Conclusions**: PCV13 vaccination influences pneumococcal serotype distribution and antimicrobial susceptibility, although there was no significant difference regarding carriage rates between vaccinated and unvaccinated groups. These findings highlight the critical importance of vaccination in reducing invasive serotypes and antimicrobial resistance in children under five years old, emphasizing the importance of national PCV vaccination programs alongside continuous serotype surveillance.

## 1. Introduction

*Streptococcus pneumoniae* (*S. pneumoniae*) continues to be a major global pathogen, causing significant illness and mortality among various age groups, particularly affecting children under five years old, who are especially at a higher risk [1]. The World Health Organization (WHO) reports that *S. pneumoniae* is responsible for approximately 300,000 deaths annually among children under the age of five worldwide. The majority of these fatalities occur in developing countries, where access to healthcare resources and preventive measures remains limited [2].

*S. pneumoniae* has the capacity to asymptomatically colonize the human nasopharynx, which promotes horizontal transmission, establishing a reservoir for community spread and invasive diseases [3]. A crucial aspect of its pathogenicity is the polysaccharide capsule, which provides immune evasion and forms the foundation for classification into more than 100 serotypes [4]. These serotypes exhibit significant differences in their geographical distribution, virulence, and patterns of antibiotic resistance, highlighting the necessity for region-specific epidemiological monitoring [5].

Antibiotic treatment is the primary approach for managing pneumococcal infections; however, the global rise in antibiotic resistance, particularly to β-lactams, poses significant challenges [6]. These trends emphasize the importance of disease prevention. Numerous studies have demonstrated that vaccination is an effective approach to reducing the global disease burden by targeting specific serotypes of *S. pneumoniae* responsible for invasive diseases such as pneumonia, meningitis, and bacteremia [7]. However, the selection pressures induced by pneumococcal conjugate vaccines (PCVs) have resulted in the rise of non-vaccine serotypes as the predominant pathogens, which are frequently linked to less severe, non-invasive infections. These serotypes play a significant role in sustaining pneumococcal colonization and transmission within communities, underscoring the potential need for new vaccine formulations that target emerging serotypes [8].

Antibiotic resistance exacerbates the challenges of managing pneumococcal infections. The prevalence of resistance to key antibiotics, including penicillin, macrolides, and cephalosporins, is on the rise, with multidrug-resistant (MDR) strains becoming an increasingly significant proportion of clinical isolates, posing an alarming global public health issue [9]. Moreover, the introduction of PCVs has revolutionized public health by markedly decreasing the incidence of antibiotic-resistant strains and diminishing antibiotic consumption through improved herd immunity [10]. This intersection of serotype replacement and antimicrobial resistance underscores the critical necessity for surveillance systems to effectively track transmission patterns and trends of antimicrobial resistance in reducing invasive pneumococcal diseases (IPDs).

The 13-valent pneumococcal conjugate vaccine (PCV13, Prevenar-13 Pfizer) is administered according to a CDC-recommended schedule in Iran [11]; however, it is administered for limited populations in private healthcare settings due to the lack of a national vaccination program against *S. pneumoniae*. The 10-valent pneumococcal conjugate vaccine (PCV10, Pneumosil, Serum Institute of India, Pune, Maharashtra, India) was introduced into the national vaccination schedule in our country in August, 2024. Therefore, there is a combination of vaccinated and unvaccinated populations included in the present research, which started before the administration of PCV10. To the best of our knowledge, this is the first investigation which determines the prevalence of nasopharyngeal colonization of *S. pneumoniae* in children aged 18 to 59 months and compares this rate between PCV13-vaccinated and -unvaccinated populations. Moreover, serotype distribution and antibiotic resistance patterns were evaluated.

## 2. Materials and Methods

### 2.1. Study Population

This multi-center, cross-sectional observational study was performed in Tehran between August 2023 and November 2024. The enrolled population included children aged from 18 to 59 months, with both PCV13-vaccinated and -unvaccinated cases with a negative history of respiratory infections during the two weeks prior to sampling. Subjects with anatomical abnormalities hindering the collection of nasopharyngeal (NP) samples were excluded.

Those eligible for the vaccinated group were those who had received their last PCV13 dose at least 6 months prior to participation.

The number of PCV13 doses varied depending on the age at the time of first vaccine dose administration: four doses for infants (aged 2–6 months), three doses for those aged 7–11 months, two doses for those aged 12–23 months, and one dose for those aged 24–59 months [11].

As PCV13 is not part of the national immunization schedule in Iran, several factors have influenced the number of doses administered to the study population, including variability in parental acceptance and compliance, limited vaccine availability, affordability, and the preference of the attending physician.

The vaccination status of the children was categorized into unvaccinated (no dose of PCV) and vaccinated (including subjects who received 1–2 doses of PCV13 and those who received ≥3 doses of PCV).

Informed written consent was obtained from parents or guardians prior to participation, and demographic information along with vaccination status was collected using standardized questionnaires administered by trained healthcare professionals during sample collection. The study protocol was approved by the ethical committee of Pasteur Institute of Iran (Ethics code IR.PII.REC.1402.015) in August 2023 and performed in compliance with the Helsinki declaration.

### 2.2. Data Collection

A structured questionnaire was employed to systematically collect the baseline demographic data and clinical characteristics of all enrolled children. Following written informed consent from parents or legal guardians, face-to-face interviews were performed by trained experts to gather standardized information on demographic variables (such as age, gender, parents’ education level, and geographic location). Individual (such as daycare attendance, underlying disease, prior hospitalizations, and history of respiratory infections) and household risk factors (such as bed sharing, exposure to cigarette smoke, and the presence of siblings) were comprehensively assessed. Respiratory infections including lower and upper respiratory tract infections were confirmed according to medical records (during one month prior to participation) by the pediatrician’s diagnosis.

### 2.3. Sample Size Calculations

The sample size was determined based on a two-sided type I error probability (*α*) of 0.05 and a statistical power of 80% (*β* = 0.20). Assuming pneumococcal carriage rates of 10.3% in the non-vaccinated group and 25.3% in the one-dose-vaccinated group [12], the required sample size per group was calculated as follows:n=z1−α2+z1−β2p11−p1+p21−p2p1−p22$$n=\frac{{\left(1.96+0.84\right)}^{2}(0.10\left(1-0.10\right)+0.25(1-0.25))}{{\left(0.25-0.1\right)}^{2}}≅97$$

To account for potential attrition or non-compliance, the sample size was increased by 5% per group, resulting in a final target of 102 participants per group.

### 2.4. Sample Collection and Culture Methods

NP samples were collected using flexible wire swabs with Dacron/Rayon tips, adhering to WHO guidelines [13]. Following collection, swabs were immediately placed in 1 mL of sterilized skimmed milk–tryptone–glucose–glycerin (STGG) transport medium. Specimens were transported on ice to the laboratory and stored at −80 °C until further processing. Pneumococcal identification followed CDC-recommended protocols [14]: briefly, 200 μL of the thawed NP-STGG sample was inoculated into 6 mL of Todd Hewitt broth supplemented with 0.5% yeast extract (THY) and 1 mL of rabbit serum. The broth was incubated for 6 h at 37 °C in a CO_2_-enriched environment. Subsequently, 10 μL of the broth was streaked onto 5% sheep blood agar plates (BAP; Oxoid Ltd., Basingstoke, United Kingdom) and incubated for 18–24 h at 37 °C under 5% CO_2_.

Initial identification of *S. pneumoniae* relied on colony morphology (α-hemolytic, mucoid colonies), Gram staining (Gram-positive cocci in pairs/chains), and optochin susceptibility testing. Isolates exhibiting inhibition zones ≥ 14 mm (optochin-sensitive) were confirmed as *S. pneumoniae*. Optochin-resistant isolates underwent bile solubility testing using 10% sodium deoxycholate (Merck, Darmstadt, Germany, pH 7.0). Strains resistant to optochin were further validated via bile solubility assays. Quality control included *S. pneumoniae* PTCC 1800 (optochin-sensitive) and *Streptococcus mitis* ATCC 6249 (optochin-resistant). Confirmed isolates were preserved at −70 °C in STGG medium for subsequent serotyping and antimicrobial susceptibility analysis.

### 2.5. DNA Extraction and Real-Time PCR

To enhance the cell lysis of *S. pneumoniae*, lysozyme was initially applied at a concentration of 1 mg/mL. Then, genomic DNA was isolated using the High Pure PCR Template Preparation Kit (Roche Molecular Diagnostics, Mannheim, Germany) in accordance with the manufacturer’s guidelines. The confirmation of *S. pneumoniae* was achieved through real-time quantitative PCR (qPCR), targeting the *lytA* gene (autolysin) [15] and the *sp2020* gene (a putative transcriptional regulator) [16]. The DNA of *S. pneumoniae* strain PTCC 1800 was used as a quality control. Primer sequences used in this study are detailed in Appendix A.

### 2.6. PCR and cpsB Serotyping

The *cpsB* gene was amplified through PCR, and the PCR products were subjected to bidirectional sequencing using the BigDye^®^ Terminator v3.1 Cycle Sequencing Kit (Applied Biosystems, Thermo Fisher Scientific, San Diego, CA, USA). Capillary electrophoresis was carried out on a 3500 Genetic Analyzer (Thermo Fisher Scientific, Waltham, MA, USA).

Consensus sequences were constructed using CLC Genomics Workbench version 5.5 (CLC bio, Aarhus, Denmark) and were further analyzed through BLAST (URL link: https://blast.ncbi.nlm.nih.gov/Blast.cgi?PRGRAM=blastn&PAGE_TYPE=BlastSearch&LINK_LOC=blasthome, accessed on 12 May 2025) against the GenBank database. Sequences that displayed a 98% similarity with entries in GenBank were assigned *cpsB* genotypes. In instances where sequence similarity was below 98%, additional steps such as re-amplification, sequencing, and BLAST analysis were performed to eliminate the possibility of detection errors.

Serotypes were classified as vaccine serotypes (VT), comprising those included in PCV13—specifically, serotypes 1, 3, 4, 5, 6A, 6B (6C/6B, 6E/6B), 7F, 9V (9V/9A), 14, 18C (18B/18C), 19A, 19F, and 23F—while serotypes not covered by PCV13 were designated as non-vaccine serotypes (NVT). Pneumococcal isolates that could not be identified through *cpsB* gene sequencing were designated as non-typeable (NT). Primer sequences utilized in this study are outlined in Appendix A [17].

### 2.7. Minimum Inhibitory Concentrations (MICs)

The minimum inhibitory concentrations (MICs) of *S. pneumoniae* isolates to five antimicrobial agents, i.e., penicillin, ceftriaxone, trimethoprim–sulfamethoxazole (SXT), erythromycin, and chloramphenicol, were evaluated using a broth microdilution method. Testing was performed in sterile 96-well microtiter plates (Nuova Aptaca, Canelli, Italy) containing Mueller–Hinton broth supplemented with 5% defibrinated horse blood. Susceptibility classifications (susceptible, intermediate, or resistant) were assigned using CLSI 2021 clinical breakpoints for *S. pneumoniae*. Additional quality control for antimicrobial susceptibility testing was performed using *S. pneumoniae* PTCC 1800, *Staphylococcus aureus* ATCC 2913, *Enterococcus faecalis* ATCC 29212, and *Escherichia coli* ATCC 25922. All isolates exhibiting resistance to three or more classes of antibiotics were classified as multidrug-resistant (MDR).

### 2.8. Statistical Analysis

The variables are described using frequency and percentage. For the age variable, descriptive statistics including mean and standard deviation are reported. Comparisons between variables between PCV13-vaccinated and -unvaccinated or carriage and non-carriage groups were assessed using the Pearson Chi-square test and Fisher–Freeman–Halton exact test/Fisher exact test. To assess the association between potential risk factors and pneumococcal carriage, we applied univariate (including each covariate separately) and multivariable (including multiple covariates simultaneously to control for potential confounders) binary logistic regression models. The vaccine coverage of PCV-13 is presented with 95% Wilson confidence intervals. A significance level of 0.05 was considered. Graphical representations were generated using Excel version 2019 and R software version 4.3.1, and data analyses were conducted with SPSS software version 26.0.

## 3. Results

### 3.1. Demographic Characteristics of the Study Population

A total of 204 nasopharyngeal (NP) samples were collected, comprising 102 samples from PCV13-vaccinated children and 102 samples from unvaccinated children. The mean age of the PCV13-vaccinated and -unvaccinated children was 36.3 ± 14.5 and 34 ± 15 months, respectively. Vaccinated children from urban areas had parents with a higher academic education level and were more likely to attend daycare than the unvaccinated children. Having siblings, the number of siblings, bed sharing, and cigarette smoke exposure were lower in vaccinated children than in unvaccinated children (Table 1 shows the demographic characteristics of the subjects). Since PCV13 administration was limited to populations in private healthcare settings, families belonging to a higher socio-economic level were expected for vaccinated children.

A comparison of socio-demographic characteristics between *S. pneumoniae* carrier and non-carrier children in both groups is shown in Appendix A.

### 3.2. Pneumococcal Prevalence and Serotype Distribution

We compared the frequencies of pneumococcal serotypes between PCV13-vaccinated and -unvaccinated children to evaluate the impact of the PCV13 vaccine on the prevalence and distribution of *S. pneumoniae* serotypes. Among the 102 PCV13-vaccinated children, 20.6% (*n* = 21) were identified as carriers of *S. pneumoniae*. Similarly, 21.6% (*n* = 22) of the unvaccinated children were found to harbor this bacterium in their nasopharynx.

In the unvaccinated group, 81.8% (*n* = 18) of isolates were VT serotypes, 13.7% (*n* = 3) were NVT serotypes, and 4.5% (*n* = 1) were NT. The identified serotypes included 19F (3), 23F (3), 14 (2), 6A (1), 6B (1), 4 (1), 18B/18C (1), 6E/6B (2), 9V/9A (2), 6C/6B (1), 19A (1), 23A (1), 15B/15C (1), NT (1), and 11A/11D/18F (1).

During the process of re-culturing for serotyping, two pneumococcal isolates from the PCV13-vaccinated group did not exhibit growth. NVT serotypes were more prevalent in the PCV13-vaccinated group compared to VT (63.2% vs. 36.8%). The identified serotypes included 15B/15C (5), 19F (4), 35F/47F (3), 9V/9A (21), 9V (1), 3 (1), 17F/33C (1), 22F/22A (1), 23A (1), and 9N/9L (1) (Figure 1).

The PCV13-vaccinated group exhibited a significantly higher prevalence of NVT and a lower prevalence of VT compared to the unvaccinated group (*p* = 0.001).

Overall, the pneumococcal carriage rate was similar between both groups; however, the specific *S. pneumoniae* serotypes differed, with a notable shift toward NVT serotypes in the vaccinated group.

### 3.3. PCV13 Coverage

Among pneumococcal isolates from PCV13-vaccinated children, seven belonged to serotypes included in PCV13, whereas eighteen isolates from unvaccinated children were of PCV13 serotypes. The vaccine coverage in the PCV13-vaccinated group was 36.8% (95% CI: 19.2–59.0%), whereas it was higher in the unvaccinated group (81.8% (95% CI: 61.5–92.7%)), indicating that PCV13 serotypes were more prevalent among unvaccinated children compared to PCV13-vaccinated children (Table 2).

### 3.4. Distribution of S. pneumoniae Serotypes in Children Based on the Number of PCV13 Doses

To investigate whether the number of PCV13 doses influences *S. pneumoniae* carriage, we analyzed the frequency of pneumococcal isolates and their serotypes according to the number of PCV13 doses administered.

Among children who had not received any PCV13 doses, the NP carriage rate of *S. pneumoniae* was 21%. In comparison, the prevalence was 28% in children who received three or more doses, and 11% in those who received one to two doses. However, these differences in overall carriage rates were not statistically significant.

Notably, the proportion of PCV13 serotypes was significantly lower in children who received three or more doses compared to those who received one to two doses (33.3% vs. 50%). Additionally, the highest prevalence of NVT serotypes was observed in children who received three or more doses (66.7%), followed by those who received one to two doses (50%) and those who received no doses (14.3%). The distribution of VT and NVT serotypes across the three groups differed significantly (*p* = 0.003).

For children who received three or more doses of the PCV13 vaccine, serotypes 15B/15C and 19F were identified as the most prevalent in NP samples. Conversely, serotypes 19F, 3, 35F/47F, and 23A were detected in children who only received two doses. Children who did not receive any doses of the PCV13 vaccine predominantly carried serotypes 23F and 19F (Figure 2). These findings suggest that the number of administered PCV13 doses might have an impact on the distribution of *S. pneumoniae* serotypes.

### 3.5. Antimicrobial Resistance Profile of S. pneumoniae Carriage Isolates

To assess the antimicrobial resistance profile of *S. pneumoniae* and identify MDR isolates among PCV13-vaccinated and -unvaccinated children, we conducted antibiotic susceptibility testing for five antibiotics.

During the process of re-culturing for the antimicrobial susceptibility test, one pneumococcal isolate from the PCV13-vaccinated group did not exhibit growth. All 20 pneumococcal serotypes isolated from this group were susceptible to penicillin, whereas 4 *S. pneumoniae* isolates (18.2%, *n* = 2 resistant, *n* = 2 intermediate) from unvaccinated children exhibited non-susceptibility to penicillin. Susceptibility to ceftriaxone was higher in the PCV13-vaccinated group compared to the unvaccinated group (90% vs. 81.8%). Additionally, both groups demonstrated high susceptibility to chloramphenicol, with the PCV13-vaccinated group showing a higher susceptibility rate than the unvaccinated group (95% vs. 86.4%). However, both PCV13-vaccinated and -unvaccinated groups exhibited significant resistance to erythromycin and SXT, with similar resistance patterns observed in both groups (Table 3). The difference in antibiotic resistance between the two groups did not show statistical significance.

Multidrug resistance was observed in 10% (*n* = 2) and 13.6% (*n* = 3) of the isolates from PCV13-vaccinated and -unvaccinated children, respectively, resulting in a total MDR rate of 11.9% in our study groups.

### 3.6. The Association of S. pneumoniae Serotypes with Antibiotic Resistance

To investigate whether the PCV13 vaccine affects the prevalence of commonly encountered antibiotic-resistant serotypes, a comparative analysis of specific resistant serotypes between PCV13-vaccinated and -unvaccinated children was performed.

Our findings revealed that serotypes 23F, 6E/6B, 6B, and NT isolated from unvaccinated children exhibited resistance to penicillin. Resistance to ceftriaxone was observed in six isolates, including serotypes 23F, 4, and NT in unvaccinated children, as well as serotypes 19F and 9N/9L in PCV13-vaccinated children.

In the unvaccinated group, serotypes 23F, 6E/6B, and NT demonstrated resistance to chloramphenicol. The most resistant isolates to SXT were serotypes 23F, 19F, and 6E/6B in unvaccinated children, and serotypes 15B/15C and 19F in vaccinated children. Serotypes 19F, 15B/15C, 23F, and 6E/6B were predominantly associated with erythromycin-resistant isolates of *S. pneumoniae*. Among erythromycin-resistant isolates, serotypes 19F and 15B/15C were more frequently observed in PCV13-vaccinated children, whereas serotypes 23F, 6E/6B, and 19F were exclusively found in the unvaccinated group.

Notably, co-resistance to both penicillin and ceftriaxone was observed in two isolates from unvaccinated children, specifically serotypes 23F and NT. Furthermore, serotypes 23F and 6E/6B in unvaccinated children, as well as serotype 19F in PCV13-vaccinated children, displayed simultaneous resistance to three or four classes of antibiotics, categorizing them as MDR strains (Appendix A).

### 3.7. Risk Factors for Pneumococcal Carriage

To identify the risk factors associated with *S. pneumoniae* NP carriage, we employed a regression model to assess the relationship between individual and household variables with carriage status. A comparative analysis of risk factors between carriers and non-carriers of *S. pneumoniae* is detailed in Table 4.

Regression analysis highlighted previous respiratory infections as a significant risk factor of pneumococcal carriage, yielding *p*-values of 0.02 in the crude analysis and 0.05 in the adjusted analysis. Moreover, daycare attendance emerged as a pronounced risk factor, elevating the likelihood of carriage by 2.95-fold in the crude model and 4.95-fold in the adjusted model (*p*-value crude: 0.024 and adjusted: 0.026). Residing in Tehran, compared to other cities, was also linked to an increased risk of colonization in the adjusted regression model (*p*-value 0.028). Additionally, antibiotic treatment within the preceding three months was associated with an increased risk of carriage, with a 2.12-fold increase (*p*-value = 0.039 in the crude analysis) (Table 4).

## 4. Discussion

Pneumococcal pneumonia remains the primary cause of mortality among children under five years old worldwide [18]. This is especially significant as *S. pneumoniae* has been identified as one of the priority pathogens by the WHO [19]. Numerous studies have shown that the rates of NP colonization by *S. pneumoniae* are notably higher in lower- and middle-income countries [20]. PCV13 vaccination was restricted to a limited population of children, resulting in insufficient data on the prevalence of pneumococcal colonization and serotype distribution among vaccinated individuals. Such targeted vaccination in a limited population may induce circulating serotype alteration, underscoring the necessity for comparative studies between vaccinated and unvaccinated individuals. For the first time, this study compared pneumococcal prevalence, serotype distribution, and antimicrobial susceptibility patterns between PCV13-vaccinated and -unvaccinated Iranian children aged 18 to 59 months.

The present data showed a similar carriage rate of *S. pneumoniae* between PCV13-vaccinated and -unvaccinated children. Among unvaccinated children, serotypes 23F and 19F were the most frequent isolates, whereas serotype 15B/15C was the most prevalent among PCV13-vaccinated children. Notably, most *S. pneumoniae* serotypes isolated from the unvaccinated group belonged to VT serotypes, while NVT serotypes were more common in the vaccinated group. Similar to our findings, a study reported a lower prevalence of NVT serotypes (16%) compared to VT serotypes in unvaccinated Iranian children [21]. Moreover, a study conducted by Gupta et al. in India reported similar NP carriage rates of *S. pneumoniae* between PCV-vaccinated and -unvaccinated children, with a higher prevalence of VT serotypes in the unvaccinated group. In their study, serotypes 23F, 19F, and 10A were predominant in the vaccinated group, while serotypes 23F, 19A, and 6A were most common in unvaccinated children [22]. A recent study reported similar pneumococcal carriage rates between children who were vaccinated with PCV10 and those who were unvaccinated. The predominant serotypes differed between groups: serotypes 28F, 23A, and 6A were most common among unvaccinated children, while serotypes 28F, 23B, 3, and 20 were more frequently observed in vaccinated children [23]. Another study also confirmed a shift toward NVT serotypes following vaccination, despite the overall carriage rate remaining unchanged [24]. Variations in serotype distribution observed across different studies may be influenced by factors such as geographic location, characteristics of the study population, vaccination coverage levels, and ethnic diversity. In general, the higher prevalence of PCV13 serotypes in unvaccinated children suggests that these serotypes continue to circulate predominantly in this group, reflecting the vaccine’s effectiveness in reducing invasive-serotype carriage among vaccinated individuals.

Numerous studies have highlighted the epidemiological impact of the COVID-19 pandemic on *S. pneumoniae* prevalence [25,26,27,28]; however, there is a lack of data regarding pneumococcal prevalence and serotype distribution in Iran following the pandemic. Our study is the first to report pneumococcal prevalence and serotype distribution in Iran following the COVID-19 pandemic. We found that the prevalence of *S. pneumoniae* colonization was 21%, with serotypes 23F and 19F being the most commonly identified among unvaccinated children. This finding is consistent with a systematic review conducted prior to the COVID-19 pandemic, which reported a 20% prevalence of *S. pneumoniae* carriage among healthy individuals, with serotypes 23F, 19F, 6A/B, 19A, and 18C being predominant in unvaccinated Iranian children [29]. Similarly, another systematic review in Iran also identified serotypes 23F and 19F as the most prevalent among unvaccinated children, underscoring the continued dominance of these serotypes in this population [30]. The present study and previous studies indicate that the prevalence and serotype distribution of pneumococcal carriage in unvaccinated children in Iran have remained largely unchanged in the post-COVID-19 era.

Monitoring the epidemiology of *S. pneumoniae* serotypes in children is crucial for assessing PCV coverage, thereby providing insights into vaccine effectiveness and guiding public health strategies [31]. Our study found that PCV13 coverage among the unvaccinated group was higher than in the PCV13-vaccinated group. Similarly, a previous study of unvaccinated six-month-old carriers in Iran estimated PCV13 coverage at 73% [32]. Moreover, a study conducted by Kielbik et al. revealed that PCV13 serotypes were more prevalent in the unvaccinated population compared to the vaccinated group, indicating a higher coverage of these serotypes in unvaccinated children [33]. Another study reported that the PCV13 coverage rate was greater among non-vaccinated Indian children than children who received PCV13 [22]. Moreover, a recent study showed high PCV13 coverage (85.2%) in unvaccinated children under 5 years [34]. These findings indicate that the administration of the PCV13 vaccine reduces invasive pneumococcal serotypes targeted by the vaccine but can lead to serotype replacement, where non-invasive or less virulent strains occupy the vacated ecological niche. These replacement strains, while generally causing less severe disease, continue to colonize the nasopharynx and facilitate transmission.

Furthermore, our findings indicate that the prevalence of PCV13 serotypes was reduced in children who received three or more doses compared to those who received only one or two doses. Among children with ≥3 doses, serotype 19F was the most commonly identified VT serotype, whereas both serotypes 19F and 3 were detected in children who had received 1 or 2 doses. Consistent with our results, previous studies have reported a significant decrease in VT serotypes in the nasopharynx of children receiving two or more doses compared to those with only a single dose [22]. Furthermore, children who were vaccinated demonstrated lower rates of pneumococcal carriage, whereas vaccinated children exhibited higher carriage rates of NVT serotypes compared to their unvaccinated counterparts [35]. These results indicate that completing vaccination series effectively reduces the carriage of VT pneumococcal serotypes.

Much epidemiological research has demonstrated that the implementation of PCVs has contributed to a general reduction in antimicrobial resistance among pneumococcal strains [36]. Our findings demonstrate differences in antibiotic susceptibility patterns between vaccinated and unvaccinated children. Vaccinated children exhibited no penicillin-resistant pneumococcal isolates, contrasting with the four non-susceptible isolates in unvaccinated children. Higher susceptibility to ceftriaxone and chloramphenicol was observed in the PCV13-vaccinated group, though both groups displayed substantial resistance to erythromycin and SXT. MDR strains were similar between PCV13-vaccinated and -unvaccinated children, with serotypes 23F, 6E/6B, and 19F driving resistance patterns. Similarly, a systematic review conducted in Iran indicated that *S. pneumoniae* isolates exhibited the highest levels of resistance to co-trimoxazole and erythromycin [29]. Similarly to our results, data from Poland revealed higher antibiotic resistance rates in *S. pneumoniae* isolates from unvaccinated children aged 1 to 6 years compared to their vaccinated counterparts [33]. Furthermore, following the introduction of the PCV, a significant reduction in antibiotic non-susceptibility among *S. pneumoniae* isolates was observed in Serbian children [37]. Collectively, these findings highlight the beneficial impact of PCVs in reducing antibiotic-resistant serotypes and IPD in children, contributing to global efforts against AMR. Nevertheless, the persistent prevalence of resistance, particularly to macrolides, along with the risk of serotype replacement, emphasizes the urgent necessity for effective antibiotic stewardship programs to maintain these gains and combat AMR trends.

The burden of NP pneumococcal colonization varies due to several factors, including the socio-demographic and clinical characteristics of the population, seasonal changes, and geographic location [38]. In our study, previous respiratory infections, daycare attendance, residence in Tehran, and recent antibiotic use were all associated with an increased risk of pneumococcal carriage. Both previous respiratory infections and daycare attendance are well-established risk factors, likely due to increased exposure to respiratory pathogens. The observed correlation between recent antibiotic use and elevated pneumococcal carriage rates may be a consequence of antibiotic-induced dysbiosis. The higher carriage rates in Tehran compared to other cities may be attributable to urban crowding. Consistent with our findings, previous studies have identified several independent determinants of *S. pneumoniae* carriage in young children. For example, research conducted in Tunisia found that factors such as age over two years, the presence of siblings, a history of respiratory tract infections, attendance at childcare facilities, and previous hospitalization were significantly associated with increased carriage rates [39]. Similarly, a study involving children in Cyprus reported that both daycare attendance and having siblings in the household substantially elevated the risk of pneumococcal colonization [40]. The difference in pneumococcal carriage risk factors observed in our study may be attributed to the higher socio-economic status of the vaccinated group, which could potentially bias the evaluation of associated risk factors. These observations underscore the multifactorial risk of pneumococcal carriage, highlighting the necessity of considering a range of demographic, clinical, and environmental risk factors when assessing the burden of NP colonization in pediatric populations.

This study has the strength of unique insights into pneumococcal serotype dynamics within Iran, where there is a combination of vaccinated and unvaccinated children. The comparison of pneumococcal carriage, serotype distribution, and antimicrobial susceptibility between PCV13-vaccinated and -unvaccinated children is a key strength. Notably, this study highlights the effectiveness of ≥3 doses of PCV13 in reducing VT serotypes. These findings underscore the importance of pneumococcal surveillance and support the development of future immunization strategies for Iran. Although employing the Quellung reaction, as a gold-standard method, could potentially enhance the accuracy of serotyping, the high costs associated with this method have limited its use for serotyping in some countries.

Despite some limitations, such as a small sample size, limited antibiotic panels, and the heterogeneity of study groups due to socio-economic factors, these data provide preliminary insights into serotype distribution and antibiotic resistance. Future longitudinal studies with larger sample sizes in a wider geographic area and including older ages, expanded antibiotic panels for susceptibility testing, and the incorporation of novel molecular methodologies could generalize the present results. Such approaches will enhance the understanding of temporal dynamics, serotype distribution, and evolving antibiotic resistance patterns.

## 5. Conclusions

This study is the first to directly compare pneumococcal carriage, serotype distribution, and antimicrobial susceptibility patterns between PCV13-vaccinated and -unvaccinated children in Iran. The research reveals that while overall carriage rates are similar between vaccinated and unvaccinated children, vaccinated children carry more NVT serotypes, and a complete PCV13 series reduces VT serotypes, suggesting dose-dependent protection. The absence of penicillin resistance in vaccinated children, in contrast to its presence in the unvaccinated group, underscores the benefits of vaccination. These results underscore the critical importance of vaccination in reducing invasive serotypes and AMR in children under 5 years old.

## Figures and Tables

**Figure 1 vaccines-13-00707-f001:**
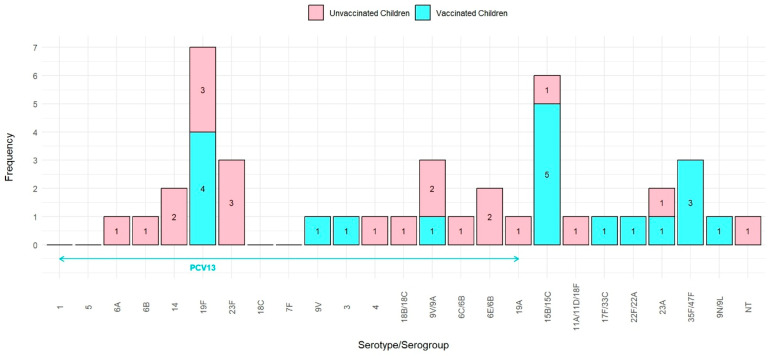
Distribution of *S. pneumoniae* serotype in PCV13-vaccinated and -unvaccinated children.

**Figure 2 vaccines-13-00707-f002:**
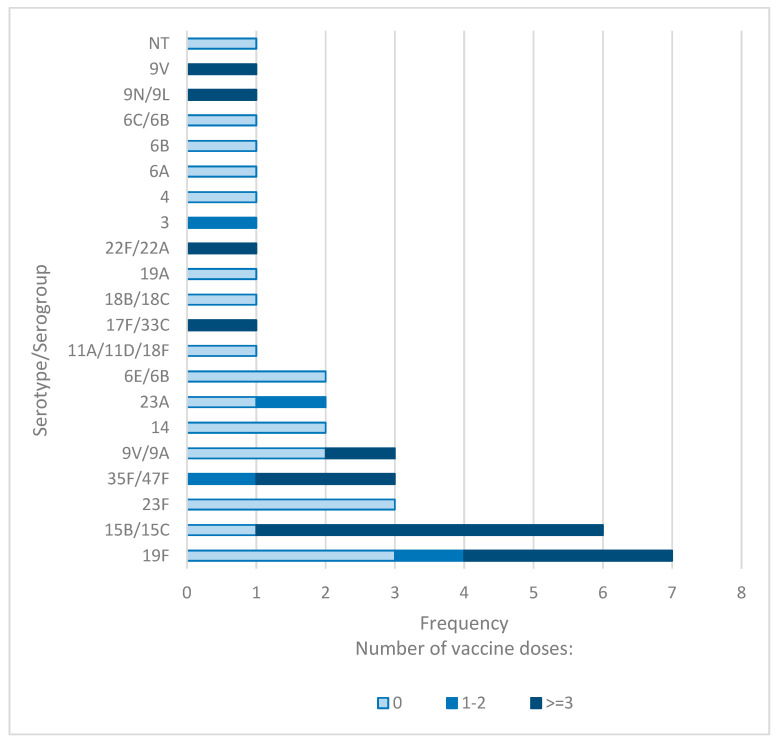
Distribution of *S. pneumoniae* serotypes in children based on PCV13 doses.

**Table 1 vaccines-13-00707-t001:** Socio-demographic characteristics of the PCV13-vaccinated and -unvaccinated children.

Variable	Category	Vaccinated (*n* (%))	Unvaccinated (*n* (%))	*p*-Value	Adjusted *p*-Value ^#^
Gender	Girl	44 (43.1)	32 (31.4)	0.082	1.000
	Boy	58 (56.9)	70 (68.6)		
Underlying disease	No	67 (65.7)	59 (57.8)	0.249	1.000
	Yes	35 (34.3)	43 (42.2)		
Geographic area	Urban	98 (96.1)	54 (57.4)	**<0.001**	**<0.001**
	Suburban/rural	4 (3.9)	40 (42.6)		
Area (Tehran)	Yes	90 (90.9)	28 (29.8)	**<0.001**	**<0.001**
	No	9 (9.1)	66 (70.2)		
Mother’s education	Academic education	60 (58.8)	29 (30.2)	**<0.001**	**0.001**
	Non-academic education	42 (41.2)	67 (69.8)		
Father’s education	Academic education	79 (80.6)	31 (32.6)	**<0.001**	**<0.001**
	Non-academic education	19 (19.4)	64 (67.4)		
Previous hospitalization	Yes	50 (49.0)	31 (30.4)	**0.007**	0.140
	No	52 (51.0)	71 (69.6)		
Cause of hospitalization	Non-infectious	28 (58.3)	20 (76.9)	0.247	1.000
	Infectious	19 (39.6)	6 (23.1)		
Antibiotic treatment within preceding 3 months	Yes	44 (43.1)	11 (10.8)	**<0.001**	**<0.001**
	No	58 (56.9)	91 (89.2)		
Previous respiratory infections	Yes	17 (16.7)	5 (4.9)	**0.007**	0.140
	No	85 (83.3)	97 (95.1)		
Type of daycare	At home	70 (72.2)	93 (95.9)	**<0.001**	**<0.001**
	Private household care	6 (6.2)	3 (3.1)		
	Daycare attendance	21 (21.6)	1 (1.0)		
Sharing bedroom with >2 persons	Yes	48 (47.1)	76 (74.5)	**<0.001**	**0.001**
	No	54 (52.9)	26 (25.5)		
Sharing bedroom with parents	Yes	46 (45.1)	72 (70.6)	**<0.001**	**0.005**
	No	56 (54.9)	30 (29.4)		
Exposure to cigarette smoke	Yes	18 (17.6)	39 (38.2)	**0.001**	**0.020**
	No	84 (82.4)	63 (61.8)		
Presence of person > 60 years old	Yes	5 (4.9)	14 (13.7)	**0.03**	0.600
	No	97 (95.1)	88 (86.3)		
Sibling	Yes	41 (40.2)	68 (66.7)	**<0.001**	**0.003**
	No	61 (59.8)	34 (33.3)		
Number of siblings	0	61 (59.8)	34 (33.3)	**<0.001** *	**0.001**
	1–2	41 (40.2)	62 (60.8)		
	≥3	0 (0.0)	6 (5.9)		
Presence of older sibling	Yes	26 (66.7)	43 (82.7)	0.077	1.000
	No	13 (33.3)	9 (17.3)		
Presence of both older and younger siblings	Yes	2 (5.1)	5 (9.6)	0.694 **	1.000
	No	37 (94.9)	47 (90.4)		
Sibling younger than 5 years	Yes	16 (40.0)	24 (46.2)	0.555	1.000
	No	24 (60.0)	28 (53.8)		

* Fisher-Freeman-Halton Exact Test. ** Fisher’s Exact Test. # *p*-values were adjusted using Bonferroni correction for multiple comparison. Bold *p*-values are shown as significant.

**Table 2 vaccines-13-00707-t002:** PCV13 coverage in PCV13-vaccinated and -unvaccinated groups.

Serogroup/Serogroup	Unvaccinated	PCV13-Vaccinated	Total
	*N* = 22	*N* = 19	*N* = 41
1	0	0	0
5	0	0	0
6A	1	0	1
6B	1	0	1
14	2	0	2
19F	3	4	7
23F	3	0	3
18C	0	0	0
7F	0	0	0
9V	0	1	1
3	0	1	1
4	1	0	1
18B/18C	1	0	1
9V/9A	2	1	3
6C/6B	1	0	1
6E/6B	2	0	2
19A	1	0	1
N	18	7	25
PCV13 coverage (95% CI)	81.8 (61.5, 92.7)	36.8 (19.2, 59.0)	61.0 (45.7, 74.3)

CI: confidence interval.

**Table 3 vaccines-13-00707-t003:** The antibiotic resistance rate among pneumococcal carriage isolates in PCV13-vaccinated and -unvaccinated children.

		Unvaccinated	Vaccinated	*p*-Value *	Total
		*N* (%)	*N* (%)		*N* (%)
Penicillin	Sensitive	18 (81.8)	20 (100.0)		38 (90.5)
	Intermediate	2 (9.1)	0 (0.0)	0.239	2 (4.8)
	Resistant	2 (9.1)	0 (0.0)		2 (4.8)
Ceftriaxone	Sensitive	18 (81.8)	18 (90.0)		36 (85.7)
	Intermediate	3 (13.6)	1 (5.0)	0.799	4 (9.5)
	Resistant	1 (4.5)	1 (5.0)		2 (4.8)
Erythromycin	Sensitive	5 (22.7)	3 (15.0)		8 (19.0)
	Intermediate	1 (4.5)	2 (10.0)	0.769	3 (7.1)
	Resistant	16 (72.7)	15 (75.0)		31 (73.8)
Chloramphenicol	Sensitive	19 (86.4)	19 (95.0)		38 (90.5)
	Intermediate	0 (0.0)	0 (0.0)	0.608	0 (0.0)
	Resistant	3 (13.6)	1 (5.0)		4 (9.5)
Trimethoprim/Sulphamethoxazole	Sensitive	5 (22.7)	4 (20.0)		9 (21.4)
	Intermediate	3 (13.6)	2 (10.0)	1.000	5 (11.9)
	Resistant	14 (63.6)	14 (70.0)		28 (66.7)

* Fisher–Freeman–Halton exact test/Fisher exact test.

**Table 4 vaccines-13-00707-t004:** Regression analysis of risk factors of *S. pneumoniae* nasopharyngeal carriage in children.

Variable	Univariate Models		Multivariable Model	
	Crude OR (95% CI)	*p*-Value	Adjusted OR (95% CI)	*p*-Value
Age (months)	1.01 (0.99, 1.03)	0.455	1.00 (0.97, 1.03)	0.938
Gender (Ref: Boy)	1	-	1	-
Girl	1.63 (0.82, 3.22)	0.16	1.52 (0.65, 3.55)	0.332
Underlying disease (Ref: No)	1	-	1	-
Yes	0.64 (0.31, 1.32)	0.226	1.04 (0.36, 2.98)	0.938
Geographic area (Ref: Urban)	1	-	1	-
Suburban/rural	1.00 (0.44, 2.31)	0.993	1.68 (0.32, 8.79)	0.536
Area (Ref: Tehran)	1	-	1	-
Other	0.53 (0.25, 1.13)	0.101	0.16 (0.03, 0.82)	**0.028**
Mother’s education (Ref: Non-academic)	1	-	1	-
Academic	0.95 (0.47, 1.89)	0.88	0.81 (0.31, 2.14)	0.672
Father’s education (Ref: Non-academic)	1	-	1	-
Academic	1.08 (0.54, 2.18)	0.822	0.77 (0.27, 2.20)	0.625
Previous hospitalization (Ref: No)	1	-	1	-
Yes	1.12 (0.57, 2.22)	0.745	1.20 (0.46, 3.09)	0.713
Antibiotic treatment within the preceding 3 months (Ref: No)	1	-	1	-
Yes	2.12 (1.04, 4.31)	**0.039**	1.28 (0.46, 3.58)	0.638
Previous respiratory infections (Ref: No)	1	-	1	-
Yes	3.01 (1.19, 7.62)	**0.02**	3.11 (1.00, 9.62)	**0.05**
Type of day care (Ref: At home)			1	-
Private household care	1.22 (0.24, 6.14)	0.812	0.83 (0.14, 5.13)	0.845
Daycare attendance	2.95 (1.16, 7.51)	**0.024**	4.49 (1.20, 16.80)	**0.026**
Sharing bedroom with more than 2 persons (Ref: No)	1	-	1	-
Yes	0.98 (0.49, 1.96)	0.962	1.34 (0.46, 3.91)	0.593
Sharing bedroom with parents (Ref: No)	1	-		
Yes	0.71 (0.36, 1.39)	0.319		
Exposure to cigarette smoke (Ref: No)	1	-	1	-
Yes	1.15 (0.55, 2.41)	0.706	1.50 (0.58, 3.86)	0.401
Presence of person > 60 years (Ref: No)	1	-	1	-
Yes	1.00 (0.31, 3.19)	0.998	0.62 (0.11, 3.38)	0.583
Sibling (Ref: No)	1	-	1	-
Yes	1.84 (0.92, 3.71)	0.086	1.53 (0.66, 3.57)	0.323
Number of siblings (Ref: None)	1	-		
1–2	1.71 (0.85, 3.44)	0.131		
>=3	1.00 (0.11, 9.14)	1.000		
Presence of older sibling (Ref: No)	1	-		
Yes	0.78 (0.30, 2.03)	0.617		
Sibling younger than 5 years (Ref: No)	1	-		
Yes	0.53 (0.20, 1.36)	0.184		
Vaccinated (Ref: No)	1	-	1	-
Yes	0.94 (0.48, 1.85)	0.864	0.42 (0.12, 1.48)	0.179
Number of vaccine doses (Ref: None)	1	-		
1–2	0.53 (0.19, 1.51)	0.235		
≥3	1.49 (0.71, 3.14)	0.294		

Bold *p*-values are shown as significant.

## Data Availability

The data that support the findings of this study are available from the corresponding authors upon request.

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
