# Peer review of "Nasopharyngeal Carriage, Serotype Distribution, and Antimicrobial Susceptibility of Streptococcus pneumoniae Among PCV13-Vaccinated and -Unvaccinated Children in Iran"

_vaccines, 2025, doi:10.3390/vaccines13070707_

Round 1
Reviewer 1 Report
Comments and Suggestions for Authors
I was invited to revise the paper entitled "Nasopharyngeal Carriage, Serotype Distribution, and Antimicrobial Susceptibility of Streptococcus pneumoniae among PCV13-Vaccinated and Unvaccinated Children in Iran". It was a cross-sectional study aimed to evaluate the prevalence of PC carriage among children from Iran.
The topic is relevant for public health and poor studies were published from this country.
Observations:
- In introduction Authors should report the vaccination schedule proposed in Iran;
- Among method, Why did Authors considered as vaccinated a patients that received only a single dose of vaccine? The schedule reports 3 doses in patients aged less than 1y, two doses in patients between 1 and 2ys and 1 dose in patients aged over 2ys;
- Sample size estimation was totally lacking;
- About statistical analysis, it is unclear the statement " and multifactorial analyses were performed using binary logistic regression". It has no sense;
- Crude analyses should be adjusted for multiple comparisons;
- Authors should analyze differences between patients with different number of vaccination received.
Author Response
Manuscript ID: vaccines-3668607
Title: Nasopharyngeal Carriage, Serotype Distribution, and Antimicrobial Susceptibility of Streptococcus pneumoniae among PCV13-Vaccinated and Unvaccinated Children in Iran
Journal name: Vaccines
Dear Editor,
On behalf of all the authors, I would like to thank you and the reviewers for their careful review of our manuscript. The suggestions were incorporated into the text of the manuscript and all the changes are marked in yellow. The answers to the comments are in blue to ease the reading.
We hope that the revised version is satisfying.
Best Regards,
Corresponding Author
Reviewer 1
Comments and Suggestions for Authors
I was invited to revise the paper entitled "Nasopharyngeal Carriage, Serotype Distribution, and Antimicrobial Susceptibility of Streptococcus pneumoniae among PCV13-Vaccinated and Unvaccinated Children in Iran". It was a cross-sectional study aimed to evaluate the prevalence of PC carriage among children from Iran.
The topic is relevant for public health and poor studies were published from this country.
Observations:
- In introduction Authors should report the vaccination schedule proposed in Iran;
Thank you for the comment. We have added the PCV13 vaccination schedule used in Iran to the Introduction section, according to CDC recommendation {four doses for infants (aged 2–6 months), three doses for 7–11 months, two doses for 12–23 months, and one dose for 24–59 months}.
- Among method, why did Authors considered as vaccinated a patients that received only a single dose of vaccine? The schedule reports 3 doses in patients aged less than 1y, two doses in patients between 1 and 2ys and 1 dose in patients aged over 2ys;
Thank you for your comment. Although the vaccine was administered according to the CDC protocol, since it is not included in the national immunization schedule, several factors have influenced the number of doses administered to children, including variability in parental acceptance and compliance, limited vaccine availability, financial issue, and the preference and opinion of the attending physician. The vaccination status of the children was categorized as unvaccinated (no dose of PCV) and vaccinated (including subjects who received 1-2 doses of PCV13 and those who received ≥3 doses of PCV).We added this classification in the method section. Therefore, similar to other studies (1-4), we included children who had received at least one dose of PCV13 as vaccinated, so that we could assess the vaccine dose potential impact on pneumococcal prevalence and serotype distribution.
- Ben Ayed N, Ktari S, Jdidi J, Gargouri O, Smaoui F, Hachicha H, Ksibi B, Mezghani S, Mnif B, Mahjoubi F, Hammami A. Nasopharyngeal Carriage of Streptococcus pneumoniae in Tunisian Healthy under-Five Children during a Three-Year Survey Period (2020 to 2022). Vaccines. 2024 Apr 9;12(4):393.
- Swarthout TD, Fronterre C, Lourenço J, Obolski U, Gori A, Bar-Zeev N, Everett D, Kamng’ona AW, Mwalukomo TS, Mataya AA, Mwansambo C. High residual carriage of vaccine-serotype Streptococcus pneumoniae after introduction of pneumococcal conjugate vaccine in Malawi. Nature communications. 2020 May 6;11(1):2222.
- Gupta P, Awasthi S, Gupta U, Verma N, Rastogi T, Pandey AK, Naziat H, Rahman H, Islam M, Saha S. Nasopharyngeal carriage of Streptococcus pneumoniae serotypes among healthy children in Northern India. Current Microbiology. 2023 Jan;80(1):41.
- Sidorenko S, Rennert W, Lobzin Y, Briko N, Kozlov R, Namazova-Baranova L, Tsvetkova I, Ageevets V, Nikitina E, Ardysheva A, Bikmieva A. Multicenter study of serotype distribution of Streptococcus pneumoniae nasopharyngeal isolates from healthy children in the Russian Federation after introduction of PCV13 into the National Vaccination Calendar. Diagnostic microbiology and infectious disease. 2020 Jan 1;96(1):114914.
- Sample size estimation was totally lacking;
Response: Thank you for your valuable comment. We added the sample size calculation in the method section.
“The sample size was determined based on a two-sided type I error probability (α) of 0.05 and a statistical power of 80% (β = 0.20). Assuming pneumococcal carriage rates of 10.3% in the non-vaccinated group and 25.3% in the one-dose vaccinated group (Chang), the required sample size per group was calculated as follows:
n=
To account for potential attrition or non-compliance, the sample size was increased by 5% per group, resulting in a final target of 102 participants per group.”
Chang B, Akeda H, Nakamura Y, Hamabata H, Ameku K, Toma T, Miyagi M, Ohnishi M. Impact of thirteen-valent pneumococcal conjugate vaccine on nasopharyngeal carriage in healthy children under 24 months in Okinawa, Japan. Journal of Infection and Chemotherapy. 2020 May 1;26(5):465-70.
- About statistical analysis, it is unclear the statement "and multifactorial analyses were performed using binary logistic regression". It has no sense;
Response: Thank you for your attention. We corrected this sentence in the method section.
“To assess the association between potential risk factors and pneumococcal carriage, we applied univariate model (including each covariate separately) and multivariable model (including multiple covariates simultaneously to control for potential confounders) binary logistic regression models.”
- Crude analyses should be adjusted for multiple comparisons;
Response: Thank you for the suggestion. We have done this evaluation and added to the data accordingly.
- Authors should analyze differences between patients with different number of vaccination received.
Response: Thank you for your comment. We analyzed and added the difference between prevalence of S. pneumoniae, VT, and NVT serotypes distribution based on vaccine dose number in the result section.
“Among children who had not received any PCV13 doses, the nasopharyngeal carriage rate of S. pneumoniae was 21%. In comparison, the prevalence was 28% in children who received three or more doses, and 11% in those who received one to two doses. However, these differences in overall carriage rates were not statistically significant. Notably, the proportion of PCV13 serotypes was significantly lower in children who received three or more doses compared to those who received one to two doses (33.3% vs. 50%). Additionally, the highest prevalence of NVT serotypes was observed in children who received three or more doses (66.7%), followed by those who received one to two doses (50%) and those who received no doses (14.3%). The distribution of VT and NVT serotypes across the three groups differed significantly (p = 0.003).”
Reviewer 2 Report
Comments and Suggestions for Authors
This study examined nasopharyngeal carriage of Streptococcus pneumoniae, serotype distribution, and antibiotic resistance in vaccinated and unvaccinated Iranian children. Although there were no significant differences in retention, PCV13 vaccination affected serotype distribution, with non-vaccinated serotypes being more common in vaccinated children. Vaccination reduced penicillin non-susceptibility. Risk factors for carriage included previous respiratory infections and preschool attendance.
There are several scientific issues in this paper that need to be improved. The authors should take appropriate action.
1.Lack of longitudinal data.
This study is described as a “multicenter cross-sectional observational study”. While this helps us to understand the current situation, it does not allow us to track changes in serotype distribution and antibiotic resistance over time within the same cohort or to establish causal relationships. This limits our ability to definitively attribute observed differences to vaccination or other time-dependent factors.
2.Potential selection bias.
The authors states, “Because PCV13 vaccination was restricted to a population attending a private health care provider, we expected families of vaccinated children to have higher socioeconomic levels”. This creates a large selection bias, as the vaccinated and unvaccinated groups may not be truly comparable outside of vaccination status, and socioeconomic factors and access to health care may confound results regarding retention rates, serotype distribution, and antibiotic resistance patterns.
3.Ambiguity regarding PCV13 doses and schedules.
The “Materials and Methods” section states that “Vaccination subjects were those who had received at least one dose of 13-valent pneumococcal conjugate vaccine (PCV-13, Prevenar 13, Pfizer) at least 6 months prior to participation.” However, the “Results” and “Discussion” sections discuss “PCV13 doses” (1-2 doses vs. 3 or more doses). Since the recommended or common PCV13 schedule in Iran is not clearly stated, it is difficult to assess whether the vaccination group was optimally protected or whether the classification of 1-2 doses vs. 3 or more doses makes clinical sense in the context of a common immunization schedule.
4.Incomplete serotype information and untypable isolates.
The results section states that there were “NT (1)” isolates in the unvaccinated group and that “pneumococcal isolates that could not be identified by cpsB gene sequencing were considered nontypeable (NT).” While it is important to recognize the presence of NT isolates, further characterization of their nontypeability Further characterization, technical issues, and new serotypes are lacking, and these may be new or emerging serotypes with different retention or resistance profiles.
5.Insufficient sample size for serotype-specific analysis and resistance.
Although the overall sample size for the study was 204 samples, the number of S. pneumoniae isolates detected (21 vaccinated and 22 unvaccinated) was relatively small. Classifying these isolates into specific serotypes or antibiotic resistance profiles (Tables 2 and 3) results in very small numbers of cells in individual serotypes or resistance patterns, making it difficult to draw statistically robust conclusions about the prevalence or resistance of individual serotypes.
6.“Overall, the specific S. pneumoniae serotypes differed between the PCV13 vaccinated and non-vaccinated groups, but there were no statistically significant differences in the overall distribution.”
This statement in the “Results” section (lines 237-239) appears to contradict the primary finding regarding serotype substitution. Perhaps because of the diversity of NVT types, there is no statistical difference in the overall distribution, but the paper clearly states that the VT serotype was more predominant in the unvaccinated group and NVT in the vaccinated group. This wording may mislead readers who expect an effect of serotype substitution.
7.Limited scope of antibiotic susceptibility testing:
This study tested susceptibility to five antimicrobial agents: penicillin, ceftriaxone, trimethoprim-sulfamethoxazole, erythromycin, and chloramphenicol. While these are important, other broader antibiotic panels commonly used for pneumococcal infections such as macrolides, tetracyclines, and fluoroquinolones (for older children/adults) should also be considered for a more comprehensive picture of the antibiotic resistance landscape.
8.Ambiguous description of “past respiratory infections.”
“Past respiratory infections” has been identified as an important risk factor. However, the paper does not define what constitutes a “past respiratory infection” (e.g., type of infection, severity, frequency, confirmation of diagnosis). This lack of specificity makes it difficult to understand the clinical significance of this risk factor and compare it to the results of other studies.
9.Generalizability of findings.
This study was conducted in a specific population in Tehran, Iran, where access to PCV13 was limited to private health care providers. Findings, especially regarding serotype distribution and coverage, may not be generalizable to regions with different vaccine introduction strategies, coverage, circulating serotypes, or socioeconomic characteristics. While the discussion partially recognizes regional differences, explicit mention should be made of their impact on broader generalizability.
Author Response
Manuscript ID: vaccines-3668607
Title: Nasopharyngeal Carriage, Serotype Distribution, and Antimicrobial Susceptibility of Streptococcus pneumoniae among PCV13-Vaccinated and Unvaccinated Children in Iran
Journal name: Vaccines
Dear Editor,
On behalf of all the authors, I would like to thank you and the reviewers for their careful review of our manuscript. The suggestions were incorporated into the text of the manuscript and all the changes are marked in yellow. The answers to the comments are in blue to ease the reading.
We hope that the revised version is satisfying.
Best Regards,
Corresponding Author
Reviewer 2
This study examined nasopharyngeal carriage of Streptococcus pneumoniae, serotype distribution, and antibiotic resistance in vaccinated and unvaccinated Iranian children. Although there were no significant differences in retention, PCV13 vaccination affected serotype distribution, with non-vaccinated serotypes being more common in vaccinated children. Vaccination reduced penicillin non-susceptibility. Risk factors for carriage included previous respiratory infections and preschool attendance.
There are several scientific issues in this paper that need to be improved. The authors should take appropriate action.
- Lack of longitudinal data.
This study is described as a “multicenter cross-sectional observational study”. While this helps us to understand the current situation, it does not allow us to track changes in serotype distribution and antibiotic resistance over time within the same cohort or to establish causal relationships. This limits our ability to definitively attribute observed differences to vaccination or other time-dependent factors.
Response: Thank you for your comment. At this stage, we aimed at evaluation of the current situation of pneumococcal serotypes/prevalence through a cross-sectional study design before PCV10 vaccine introduction to the society. Therefore, tracking of the serotypes distribution was not targeted at this preliminary study. The present data could be expanded through longitudinal study to assess the trend of circulating serotypes, which has been added as a recommendation at the end of the manuscript.
- Potential selection bias.
The authors states, “Because PCV13 vaccination was restricted to a population attending a private health care provider, we expected families of vaccinated children to have higher socioeconomic levels”. This creates a large selection bias, as the vaccinated and unvaccinated groups may not be truly comparable outside of vaccination status, and socioeconomic factors and access to health care may confound results regarding retention rates, serotype distribution, and antibiotic resistance patterns.
Response: We appreciate the reviewer’s insightful comment. This comment is accurate; however, it should be noted that the data were directly derived from the real-world situation in our country. If there was any potential bias, we had no role in creating it. Our objective was solely to depict the actual circumstances prior to the implementation of the national vaccination program.
- Ambiguity regarding PCV13 doses and schedules.
The “Materials and Methods” section states that “Vaccination subjects were those who had received at least one dose of 13-valent pneumococcal conjugate vaccine (PCV-13, Prevenar 13, Pfizer) at least 6 months prior to participation.” However, the “Results” and “Discussion” sections discuss “PCV13 doses” (1-2 doses vs. 3 or more doses). Since the recommended or common PCV13 schedule in Iran is not clearly stated, it is difficult to assess whether the vaccination group was optimally protected or whether the classification of 1-2 doses vs. 3 or more doses makes clinical sense in the context of a common immunization schedule.
Thank you for your comment. Although the vaccine was administered according to the CDC protocol, since it is not included in the national immunization schedule, several factors have influenced the number of doses administered to children, including variability in parental acceptance and compliance, limited vaccine availability, financial issue, and the preference and opinion of the attending physician. The vaccination status of the children was categorized as unvaccinated (no dose of PCV) and vaccinated (including subjects who received 1-2 doses of PCV13 and those who received ≥3 doses of PCV). We added this classification in the method section. Therefore, similar to other studies (1-4), we included children who had received at least one dose of PCV13 as vaccinated, so that we could assess the vaccine dose potential impact on pneumococcal prevalence and serotype distribution.
- Ben Ayed N, Ktari S, Jdidi J, Gargouri O, Smaoui F, Hachicha H, Ksibi B, Mezghani S, Mnif B, Mahjoubi F, Hammami A. Nasopharyngeal Carriage of Streptococcus pneumoniae in Tunisian Healthy under-Five Children during a Three-Year Survey Period (2020 to 2022). Vaccines. 2024 Apr 9;12(4):393.
- Swarthout TD, Fronterre C, Lourenço J, Obolski U, Gori A, Bar-Zeev N, Everett D, Kamng’ona AW, Mwalukomo TS, Mataya AA, Mwansambo C. High residual carriage of vaccine-serotype Streptococcus pneumoniae after introduction of pneumococcal conjugate vaccine in Malawi. Nature communications. 2020 May 6;11(1):2222.
- Gupta P, Awasthi S, Gupta U, Verma N, Rastogi T, Pandey AK, Naziat H, Rahman H, Islam M, Saha S. Nasopharyngeal carriage of Streptococcus pneumoniae serotypes among healthy children in Northern India. Current Microbiology. 2023 Jan;80(1):41.
- Sidorenko S, Rennert W, Lobzin Y, Briko N, Kozlov R, Namazova-Baranova L, Tsvetkova I, Ageevets V, Nikitina E, Ardysheva A, Bikmieva A. Multicenter study of serotype distribution of Streptococcus pneumoniae nasopharyngeal isolates from healthy children in the Russian Federation after introduction of PCV13 into the National Vaccination Calendar. Diagnostic microbiology and infectious disease. 2020 Jan 1;96(1):114914.
- Incomplete serotype information and untypable isolates.
The results section states that there were “NT (1)” isolates in the unvaccinated group and that “pneumococcal isolates that could not be identified by cpsB gene sequencing were considered nontypeable (NT).” While it is important to recognize the presence of NT isolates, further characterization of their nontypeability. Further characterization, technical issues, and new serotypes are lacking, and these may be new or emerging serotypes with different retention or resistance profiles.
Response: Thank you for your comment. cpsB sequencing is unable to identify some serotypes because of genetic variations, such as point mutations, high sequence similarity among certain serotypes, and the absence or divergence of the cpsB gene in some strains. Combining serological and molecular methods could improve the accurate identification of serotypes, including novel ones. However, due to methodological constraints and limited resources, we were only able to perform cpsB gene sequencing for serotype identification. We acknowledge that incorporating additional methodologies could accurately identify NT isolate; however, such approaches were beyond the scope of the present study.
- Insufficient sample size for serotype-specific analysis and resistance.
Although the overall sample size for the study was 204 samples, the number of S. pneumoniae isolates detected (21 vaccinated and 22 unvaccinated) was relatively small. Classifying these isolates into specific serotypes or antibiotic resistance profiles (Tables 2 and 3) results in very small numbers of cells in individual serotypes or resistance patterns, making it difficult to draw statistically robust conclusions about the prevalence or resistance of individual serotypes.
Response: We acknowledge that the limited number of isolates per group reduces the statistical robustness of serotype-specific conclusions. However, despite this limitation, the data provide valuable preliminary insights into serotype distribution and antibiotic resistance within our population prior to the introduction of PCV10 into the national vaccination program. Furthermore, we recommend that larger multicenter studies to extend these findings.
- “Overall, the specific S. pneumoniae serotypes differed between the PCV13 vaccinated and non-vaccinated groups, but there were no statistically significant differences in the overall distribution.”
This statement in the “Results” section (lines 237-239) appears to contradict the primary finding regarding serotype substitution. Perhaps because of the diversity of NVT types, there is no statistical difference in the overall distribution, but the paper clearly states that the VT serotype was more predominant in the unvaccinated group and NVT in the vaccinated group. This wording may mislead readers who expect an effect of serotype substitution.
Response: Thank you for highlighting this. Although the specific types of S. pneumoniae serotypes varied between the PCV13 vaccinated and unvaccinated groups, there was no statistically significant difference in the overall S. pneumoniae prevalence between them.
- Limited scope of antibiotic susceptibility testing:
This study tested susceptibility to five antimicrobial agents: penicillin, ceftriaxone, trimethoprim-sulfamethoxazole, erythromycin, and chloramphenicol. While these are important, other broader antibiotic panels commonly used for pneumococcal infections such as macrolides, tetracyclines, and fluoroquinolones (for older children/adults) should also be considered for a more comprehensive picture of the antibiotic resistance landscape.
Response: Thank you for the great comment. As tetracyclines and fluoroquinolones are not recommended or prescribed for children under 5 years of age, they were not included in the design of this study and were therefore not evaluated. Future studies could expand the antibiotic panel to include a broader range of agents for use in older age groups.
- Ambiguous description of “past respiratory infections.”
“Past respiratory infections” has been identified as an important risk factor. However, the paper does not define what constitutes a “past respiratory infection” (e.g., type of infection, severity, frequency, confirmation of diagnosis). This lack of specificity makes it difficult to understand the clinical significance of this risk factor and compare it to the results of other studies.
Response: thank you for your suggestion. This statement “Respiratory infections including lower and upper respiratory tract infections were confirmed according to the medical records (during one month prior to participation) by the pediatrician’s diagnosis” has been added to the method section.
- Generalizability of findings.
This study was conducted in a specific population in Tehran, Iran, where access to PCV13 was limited to private health care providers. Findings, especially regarding serotype distribution and coverage, may not be generalizable to regions with different vaccine introduction strategies, coverage, circulating serotypes, or socioeconomic characteristics. While the discussion partially recognizes regional differences, explicit mention should be made of their impact on broader generalizability.
Response: Thank you for valuable comment. In fact, this study was done as a cross-sectional approach to achieve a preliminary perspective of the current situation in Iran prior to the introduction of PCV10 into the national vaccination program. Despite to the mentioned limitation, the data provide valuable preliminary insights into serotype distribution and antibiotic resistance within our population prior to the national vaccination program.
Future longitudinal studies with larger sample size in general population could generalize the present results. Such approaches will enhance understanding of temporal dynamics, serotype distribution, and evolving antibiotic resistance patterns.
Round 2
Reviewer 1 Report
Comments and Suggestions for Authors
The paper is now acceptable
Author Response
Thank you
Reviewer 2 Report
Comments and Suggestions for Authors
The reviewers' comments regarding any unresolved aspects of the authors' responses are listed below, and unless these are resolved, I will not agree to publication.
No.2 Author's Response: We appreciate the reviewer’s insightful comment. This comment is accurate; however, it should be noted that the data were directly derived from the real-world situation in our country. If there was any potential bias, we had no role in creating it. Our objective was solely to depict the actual circumstances prior to the implementation of the national vaccination program.
Reviewer’s comments: Authors accurately acknowledge the presence of selection bias due to the real-world circumstances of vaccine access in Iran. Authors explain that your objective was to depict the actual situation, which is a valid research goal. However, while authors state they had "no role in creating it," the response doesn't explicitly mention how this acknowledged bias will be addressed in the interpretation of results or discussed as a significant limitation in the manuscript itself. Simply stating the bias exists and was unavoidable without discussing its implications for the findings, or how it might have been statistically accounted for, leaves the problem unresolved from a scientific rigor perspective.
No.5 Author’s Response: Your Response: We acknowledge that the limited number of isolates per group reduces the statistical robustness of serotype-specific conclusions. However, despite this limitation, the data provide valuable preliminary insights into serotype distribution and antibiotic resistance within our population prior to the introduction of PCV10 into the national vaccination program. Furthermore, we recommend that larger multicenter studies to extend these findings.
Reviewer’s comments:The small sample size of approximately 20 strains of pneumococcus makes it impossible to evaluate the results of the study properly. Wouldn't it be necessary to present results using a sufficient number of isolated and examined pneumococci through additional research?
No.6 Author's Response: Thank you for highlighting this. Although the specific types of S. pneumoniae serotypes varied between the PCV13 vaccinated and unvaccinated groups, there was no statistically significant difference in the overall S. pneumoniae prevalence between them.
Reviewer’s comments: While the author's response clarifies that the "no statistically significant difference" refers to the overall S. pneumoniae prevalence (carriage rates), it doesn't fully address the core issue of the wording in the original text (lines 237-239) potentially misleading readers regarding serotype distribution. The problem highlighted the contradiction between the phrasing "no statistically significant difference in their overall distribution" (referring to serotypes) and the evident shift in specific serotypes (VT vs. NVT). The author's current response reiterates that there was no significant difference in prevalence, but doesn't explicitly state that the manuscript's wording in that specific line will be modified to clearly distinguish between overall carriage rates and the observed shift in serotypes.
No. 7 Author’s Response: Thank you for the great comment. As tetracyclines and fluoroquinolones are not recommended or prescribed for children under 5 years of age, they were not included in the design of this study and were therefore not evaluated. Future studies could expand the antibiotic panel to include a broader range of agents for use in older age groups.
Reviewer’s comments: Chloramphenicol is known to cause Gray's syndrome in newborns and should not be recommended for the treatment of pneumococcus, yet it is included in the drugs reviewed by the authors in this study, which contradicts the authors' response.
No.8 Author’s Response: Thank you for your suggestion. This statement “Respiratory infections including lower and upper respiratory tract infections were confirmed according to the medical records (during one month prior to participation) by the pediatrician’s diagnosis” has been added to the method section.
Reviewer’s comments: The reviewer is unable to ascertain the specific definition of "past respiratory infection."
Author Response
Manuscript ID: vaccines-3668607
Title: Nasopharyngeal Carriage, Serotype Distribution, and Antimicrobial Susceptibility of Streptococcus pneumoniae among PCV13-Vaccinated and Unvaccinated Children in Iran
Journal name: Vaccines
Dear Editor,
On behalf of all the authors, I would like to thank you and the reviewer for its careful review of our manuscript. The suggestions were incorporated into the text of the manuscript and all the changes are marked in yellow. The answers to the comments are in blue to ease the reading.
We hope that the revised version is satisfying.
Best Regards,
Corresponding Author
Reviewer 2
Comments and Suggestions for Authors
The reviewers' comments regarding any unresolved aspects of the authors' responses are listed below, and unless these are resolved, I will not agree to publication.
No.2 Author's Response: We appreciate the reviewer’s insightful comment. This comment is accurate; however, it should be noted that the data were directly derived from the real-world situation in our country. If there was any potential bias, we had no role in creating it. Our objective was solely to depict the actual circumstances prior to the implementation of the national vaccination program.
Reviewer’s comments: Authors accurately acknowledge the presence of selection bias due to the real-world circumstances of vaccine access in Iran. Authors explain that your objective was to depict the actual situation, which is a valid research goal. However, while authors state they had "no role in creating it," the response doesn't explicitly mention how this acknowledged bias will be addressed in the interpretation of results or discussed as a significant limitation in the manuscript itself. Simply stating the bias exists and was unavoidable without discussing its implications for the findings, or how it might have been statistically accounted for, leaves the problem unresolved from a scientific rigor perspective.
Response: We appreciate the reviewer’s comment. As you have also mentioned, the data were the reflection of the real situation in our country. The possible bias of socio-economic level mostly affected the risk factors. This issue has been added to the discussion section.
“The difference in pneumococcal carriage risk factors observed in our study may be attributed to the higher socio-economic status of the vaccinated group, which could potentially bias the evaluation of associated risk factors. ”
No.5 Author’s Response: Your Response: We acknowledge that the limited number of isolates per group reduces the statistical robustness of serotype-specific conclusions. However, despite this limitation, the data provide valuable preliminary insights into serotype distribution and antibiotic resistance within our population prior to the introduction of PCV10 into the national vaccination program. Furthermore, we recommend that larger multicenter studies to extend these findings.
Reviewer’s comments:The small sample size of approximately 20 strains of pneumococcus makes it impossible to evaluate the results of the study properly. Wouldn't it be necessary to present results using a sufficient number of isolated and examined pneumococci through additional research?
Response: Thank you for your comment. In the method section, we have clarified that this study was conducted prior to the introduction of the PCV10. The sample size calculation was added in the method section. According to limited time and financial source, the study population was calculated as the minimum required sample size to achieve the study objectives before PCV10 mass vaccination in Iran. It is acknowledged that a larger sample size could have been analyzed if additional time and resources were available. This limitation has been addressed in the limitations section
No.6 Author's Response: Thank you for highlighting this. Although the specific types of S. pneumoniae serotypes varied between the PCV13 vaccinated and unvaccinated groups, there was no statistically significant difference in the overall S. pneumoniae prevalence between them.
Reviewer’s comments: While the author's response clarifies that the "no statistically significant difference" refers to the overall S. pneumoniae prevalence (carriage rates), it doesn't fully address the core issue of the wording in the original text (lines 237-239) potentially misleading readers regarding serotype distribution. The problem highlighted the contradiction between the phrasing "no statistically significant difference in their overall distribution" (referring to serotypes) and the evident shift in specific serotypes (VT vs. NVT). The author's current response reiterates that there was no significant difference in prevalence, but doesn't explicitly state that the manuscript's wording in that specific line will be modified to clearly distinguish between overall carriage rates and the observed shift in serotypes.
Response: Thank you for your comment. This sentence was edited in the result section.
“Overall, the pneumococcal carriage rate was similar between both groups; however, the specific S. pneumoniae serotypes differed, with a notable shift toward NVT serotypes in the vaccinated group.”
No. 7 Author’s Response: Thank you for the great comment. As tetracyclines and fluoroquinolones are not recommended or prescribed for children under 5 years of age, they were not included in the design of this study and were therefore not evaluated. Future studies could expand the antibiotic panel to include a broader range of agents for use in older age groups.
Reviewer’s comments: Chloramphenicol is known to cause Gray's syndrome in newborns and should not be recommended for the treatment of pneumococcus, yet it is included in the drugs reviewed by the authors in this study, which contradicts the authors' response.
Response: Thank you for your comment. The study population did not include newborns (younger than 1 month), as the age criteria was set for older than 18 months. Gray syndrome is a serious and potentially fatal reaction to chloramphenicol, particularly in newborns and infants, due to their immature liver function and inability to properly metabolize the drug. Significant risk: Infants up to about 1 month of age due to immature liver enzyme systems. This age category was not included in this study.
No.8 Author’s Response: Thank you for your suggestion. This statement “Respiratory infections including lower and upper respiratory tract infections were confirmed according to the medical records (during one month prior to participation) by the pediatrician’s diagnosis” has been added to the method section.
Reviewer’s comments: The reviewer is unable to ascertain the specific definition of "past respiratory infection."
Response: According to the study concept, the centers which were in collaboration with us, provided case visiting, sampling and data. The past history of any respiratory infections might recorded in other centers, hospitals and private settings in which the diagnosis and tests were confirmed. Therefore, we were not able to access all the data in details.
Round 3
Reviewer 2 Report
Comments and Suggestions for Authors
The authors have generally responded to the reviewers' comments.